# Influence of Coronary Artery Bypass Grafts on Blood Aminothiols in Patients with Coronary Artery Disease

**DOI:** 10.3390/metabo13060743

**Published:** 2023-06-10

**Authors:** Alexander Vladimirovich Ivanov, Mikhail Aleksandrovich Popov, Arkady Andreevich Metelkin, Valery Vasil’evich Aleksandrin, Evgeniy Gennad’evich Agafonov, Maria Petrovna Kruglova, Ekaterina Vladimirovna Silina, Victor Aleksandrovich Stupin, Ruslan Andreevich Maslennikov, Aslan Amirkhanovich Kubatiev

**Affiliations:** 1Institute of General Pathology and Pathophysiology, Baltiyskaya St., 8, 125315 Moscow, Russia; armetelkin@gmail.com (A.A.M.); aleksandrin-54@mail.ru (V.V.A.); marykruglova@live.ru (M.P.K.); niiopp@mail.ru (A.A.K.); 2Moscow Regional Research and Clinical Institute n.a. M.F. Vladimirskiy, Shchepkin St., 61/2, 129110 Moscow, Russia; popovcardio88@mail.ru (M.A.P.); chafel@mail.ru (E.G.A.); rusmaslennikov@mail.ru (R.A.M.); 3Department of Human Pathology, I.M. Sechenov First Moscow State Medical University (Sechenov University), Trubetskaya St., 8, 119991 Moscow, Russia; silinaekaterina@mail.ru; 4Department of Hospital Surgery No. 1, Pirogov Russian National Research Medical University, Ostrovityanova St., 1, 117997 Moscow, Russia; stvictor@bk.ru

**Keywords:** aminothiols, coronary artery disease, coronary artery bypass graft, cysteine, glutathione

## Abstract

Coronary artery disease (CAD) and the coronary artery bypass graft (CABG) are associated with a decreased blood glutathione (bGSH) level. Since GSH metabolism is closely related to other aminothiols (homocysteine and cysteine) and glucose, the aim of this study was to reveal the associations of bGSH with glucose and plasma aminothiols in CAD patients (N = 35) before CABG and in the early postoperative period. Forty-three volunteers with no history of cardiovascular disease formed the control group. bGSH and its redox status were significantly lower in CAD patients at admission. CABG had no significant effect on these parameters, with the exception of an increase in the bGSH/hemoglobin ratio. At admission, CAD patients were characterized by negative associations of homocysteine and cysteine with bGSH. All these associations disappeared after CABG. An association was found between an increase in oxidized GSH in the blood in the postoperative period and fasting glucose levels. Thus, CAD is associated with the depletion of the intracellular pool and the redox status of bGSH, in which hyperhomocysteinemia and a decrease in the bioavailability of the extracellular pool of cysteine play a role. The present study indicates that CABG causes disruptions in aminothiol metabolism and induces the synthesis of bGSH. Moreover, glucose becomes an important factor in the dysregulation of GSH metabolism in CABG.

## 1. Introduction

Coronary artery disease (CAD) causes angina pectoris, myocardial infarction, and ischemic heart failure and thereby contributes significantly to cardiovascular disease, the leading cause of death worldwide [1]. CAD is characterized by the development of atherosclerotic plaques inside the coronary vessel wall that stenose the vessel (causing ischemia) [2]. Treatment of CAD includes two relatively invasive strategies that aim to re-establish an adequate blood supply to undersupplied myocardial territories due to severe coronary stenosis or vessel occlusion: percutaneous coronary intervention (PCI) and coronary artery bypass grafting (CABG) [1].

According to many clinical studies, increased oxidative stress (OS) and decreased antioxidant status were observed in patients with CAD [3,4,5]. Carrying out CABG, especially with the use of cardiopulmonary bypass (the so-called on-pump), as well as other operations associated with the ischemia–reperfusion of large organs, is accompanied by the activation of ROS-generating enzymes and therefore is a trigger for additional OS tissue damage [6,7]. Using the off-pump CABG technique can reduce the severity of reactive OS but does not completely suppress it [6]. In the pathogenesis of postoperative complications of CABG, such as atrial fibrillation, shunt thrombosis, diabetes mellitus (DM), and renal lung diseases, OS plays an important role [8,9]. Therefore, the study of various aspects of OS here can help minimize the risks of CABG complications and identify new targets for the medical preparation of patients for this operation.

Aminothiols, such as glutathione (γ-glutamylcysteinyl-glycine tripeptide (GSH)) and other low-molecular-weight thiols (cysteine (Cys), homocysteine (Hcy), cysteinylglycine (CG), and others), play an important role in the protection of cells and tissues from OS [10]. GSH, a major thiol antioxidant and redox buffer inside cells, is involved in redox regulation, gene expression, signal transduction, and apoptosis [11]. Its intracellular concentration is very high (about 1 mM or more); moreover, ~99% of GSH is accounted for by the reduced form of GSH (bGSH). A small proportion (<1%) of GSH is oxidized to disulfide (GSSG). The ratio between bGSH and GSSG (redox status (RS)) is a recognized index of OS [12].

GSH metabolism is influenced by other aminothiols, primarily Cys, which is the rate-limiting substrate for GSH synthesis. [13]. The main pool of Cys is found in the plasma, where its concentration is quite high (~150–300 μM) [14]. In addition, about half of Cys is formed from dietary methionine through a series of biochemical reactions, including the formation of S-adenosylmethionine (SAM), S-adenosylhomocysteine (SAH), and Hcy. Glucose also plays an important role in GSH metabolism. A negative, although not very strong, association between fasting glucose and plasma GSH was found in CAD patients but not in non-CAD controls [15]. The inhibition of glucose transport alleviated damage to cardiac function in vivo in a diabetic cardiomyopathy model and increased the level of GSH in cells [16].

Some aminothiols are associated with CAD. Therefore, numerous clinical studies have reported associations of hyperhomocysteinemia (HHcy) with severity of CAD and risk of complications, morbidity, and mortality after CABG or stenting in patients with heart failure [5,15,17,18,19,20]. An increase in Hcy was found in the postoperative period of CABG (1–6 weeks); moreover, this was not due to such “trivial” Hcy regulation factors as decreases in renal function or vitamin B9 levels [21]. At the same time, the presence of a close association between levels of tHcy and troponin T in the CABG postoperative period indicates that Hcy is a factor in or marker of myocardial damage [22].

A number of studies have identified an association of high levels of Cys with cardiovascular disease [23], as well as an association of cystine (Cys disulfide) with cardiovascular disease mortality [24]. Many clinical studies have demonstrated associations of low levels of bGSH, the bGSH/hemoglobin (Hb) ratio [3,4], plasma total GSH (tGSH) [5], and reduced GSH [25] with CAD and heart failure. Additionally, a decrease in RS GSH in the blood of patients with indications for heart surgery (CABG and aortic or mitral valve transplantation) compared with healthy controls was found in [26]. The association of low reduced GSH with high cystine levels in plasma as well as their relationship with mortality in patients with CAD, independent of inflammatory burden, were identified in studies [27,28]. However, it is not clear whether this simply reflects a decrease in RS of plasma aminothiols or is associated with suppression of GSH synthesis due to inhibition of Cys transport into cells.

GSH, as the main intracellular antioxidant, has long attracted attention as a target of metabolic therapy [29], since normalization of its level can reduce the risk of postoperative complications in CABG. In this regard, it is interesting to identify the factors that affect GSH and, accordingly, to identify groups of patients with the highest risk of its metabolism disorders in the postoperative period.

It is known that even short-term blood loss can be accompanied by significant changes in the plasma aminothiol system [30]. CABG, in turn, can be considered a procedure that triggers stress response and adaptive mechanisms. Currently, there is still insufficient information about the features of the aminothiol system in patients with CAD and about changes in this system in the postoperative period after CABG. In particular, nothing is known about the effect CABG has on aminothiols, primarily GSH, in the blood in the early postoperative (4–5 days) period, and whether glucose levels may play a role here. It is also not known which associations of aminothiols with each other are characteristic of CAD and whether they are retained after CABG. To shed light on these questions, we conducted a study of the total contents of total plasma aminothiols (tCys, tHcy, tCG, tGSH), Hcy precursors (SAM, and SAH) and reduced and oxidized GSH in the blood (bGSH and GSSG, respectively) of patients in the preoperative and early postoperative (4–5 days) CABG periods.

## 2. Materials and Methods

### 2.1. Patient Characteristics

This study was conducted from 1 March to 30 June 2022 in accordance with the ethical principles of the Declaration of Helsinki of the World Medical Association (1964 and 2004) and the written voluntary informed consent of all patients. The study was approved by the ethics committee of the Moscow Regional Research Clinical Institute (protocol No. 3 from 3 March 2022).

The study included patients suffering from ischemic heart disease before and after surgical myocardial revascularization. The study included 35 patients with ischemic heart disease, the average age of whom was 55.4 ± 9.6 years. The main inclusion criteria for patients in the main group were the presence of hemodynamically significant stenosis of the main coronary arteries (the anterior interventricular artery, circumflex artery, and right coronary artery), planned operation (surgical myocardial revascularization), an age of 45–75 years, and the presence of informed consent. Exclusion criteria included acute myocardial infarction, chronic heart failure (functional classes III–IV), oncological diseases, and blood diseases.

Most of the patients had multivessel coronary artery disease. All patients underwent an examination that included an assessment of the physical condition, clinical and biochemical analyses, electrocardiography, and echocardiography. Diagnosis of coronary artery disease was confirmed using coronary angiography. All patients underwent surgical myocardial revascularization (CABG). Median sternotomy was used as a standard surgical approach. Artificial blood circulation was performed (right atrium–ascending aorta) in normothermic conditions. The internal thoracic artery was used as a bypass of the anterior interventricular artery. Autovenous grafts were used for the remaining coronary arteries. The operation ended according to the standard protocol. Immediately before surgical treatment, all patients stopped taking antiplatelet agents, and the administration of these agents resumed in the early postoperative period, in accordance with clinical recommendations. In all patients, the postoperative period was uneventful, and discharge from the clinic occurred on day 7–10.

The control group consisted of 43 volunteers aged 40–75 years. The inclusion criterion for this cohort was the absence of cardiovascular pathology, blood diseases, kidney diseases, a history of oncological diseases, and the use of narcotic drugs.

### 2.2. Laboratory Studies

Patients’ fasting venous blood was collected in 3 mL K3EDTA tubes (Lab-Vac, Heze, China) at admission and 4–5 days after CABG. Blood samples for the study of hemostasis parameters were obtained by venipuncture from the ulnar veins (~12 mL) using disposable vacuum systems containing 2.5% sodium citrate. To study the biochemical parameters, blood was taken into a test tube with a coagulation activator. Samples were processed no later than 30 min after venipuncture. An ACL TOP 700 hemostasis analyzer (IL Werfen, Barcelona, Spain), an AU 680 biochemical analyzer (Beckman Coulter, Brea, CA, USA), and a PENTRA 120 hematology analyzer (Horiba ABX, Montpellier, France) were used.

Determination of IL-6 in plasma was carried out using test systems from Bender Medsystems GmbH (Vienna, Austria) according to the manufacturer’s instructions [31].

### 2.3. Determination of bGSH, GSSG, and RS GSH

Blood samples (3 mL) were immediately mixed with 350 μL of 0.5 M Na citrate (pH 4.3) for determination of aminothiols in blood or plasma. Samples were cooled at 4 °C for 1.5–2 h. To obtain plasma, cooled blood was centrifuged for 3 min at 4000× *g*. Blood GSH, GSSG, and RS GSH (2·bGSH/GSSG) were determined via capillary electrophoresis, as previously described [32]. The precision of the analysis was within 3.5%, and the correctness was 101–105%.

### 2.4. Determination of Aminothiols in Plasma

Determination of aminothiols in plasma (tCys, tCG, tGSH, and tHcy) was carried out using liquid chromatography. First, 10 µL of 0.1 M dithiothreitol, 10 µL of 0.5 mM penicillamine, 10 µL of 0.4 M Na-phosphate buffer (pH 8.0), 8 µL of 1 M NaOH, and 12 µL of water were added to 50 µL of plasma, and the mixture was incubated for 30 min at 37 °C. Then, 300 μL of 67 mM 5,5′-dithiobis-(2-nitrobenzoic acid) in acetonitrile with 10% ethanol was added, and after stirring and centrifugation (5 min at 15,000× *g*), 10 μL of 1 M HCl and 150 μL of CHCl3 were added to 300 µL of the supernatant. After vigorous stirring, the mixture was centrifuged for 2 min at 3000× *g* and the upper phase was taken, which was dried under vacuum (30 min at 45 °C). The residue was redissolved in 200 µL of water. Then, 10 µL of the sample was injected into the chromatograph (Acquity UPLC H-class, Waters, Milford, MA, USA). Chromatography was performed using a Zorbax Eclipse plus C18 Rapid Resolution HD column (150 mm × 2.1 mm × 1.8 μm; Agilent, Santa Clara, CA, USA) with a gradient of acetonitrile from 2.5% to 14% in the presence of 0.15 M NH_4_ acetate with 0.1% formic acid for 7 min at a flow of 0.15 mL/min (column temperature: 40 °C), followed by a column wash with 50% acetonitrile (1 min) and 2.5% acetonitrile (6 min). The signal was detected using an absorption level at 330 nm. The precision of the analysis was within 5%, and the correctness was 93–104%.

### 2.5. Determination of SAM and SAH in Plasma

Determination of SAM and SAH in plasma was carried out using liquid chromatography with fluorescence detection, as described in [33], with slight changes in sample preparation. An extraction cartridge containing 10 mg of the Bond Elut PBA phase (Agilent, Santa Clara, CA, USA) was loaded with 200 µL of 0.1 M Na-phosphate buffer (pH 8.0), 15 µL of 2.5 µM S-(5′-Adenosyl)-3-thiopropylamine (internal standard), 25 µL of 1 M NaOH, and 150 µL of plasma. The mixture was stirred rapidly, and the cartridge was washed with 0.8 mL of 10 mM Na-phosphate buffer (pH 7.0). The analytes were eluted with 0.1 mL of 250 mM HCl. Derivatization of the analytes was carried out for 4 h at 37 °C by adding 37 µL of 1 M Na acetate (pH 5.0), 18 µL of 1 M NaOH, and 30 µL of 50% choloroacetaldehyde to the eluate, followed by the addition of 7.5 µL of formic acid to stop the reaction. The precision of the analysis was within 9%, and the correctness was 97–101%.

### 2.6. Data Processing

Data collection and primary processing (identification and integration of the chromatographic peaks) were performed using MassLynx v4.1 (Waters, Milford, MA, USA) and Elforun software v. 4.2.5 (Lumex, St. Petersburg, Russia). Statistical data analysis was performed using SPSS Statistics v. 22 (IBM, Armonk, NY, USA). Quantitative indicators were expressed as medians (and 1st and 3rd quartiles). Nonparametric Wilcoxon and Mann–Whitney tests with Holm–Bonferroni corrections for multiple-group comparisons and Pearson’s or Spearman’s rank correlation coefficient (ρ) were used to compare groups on a quantitative basis. Comparison of binomial indicators (variable analysis) was carried out via the relative risk ratio (RR); *p* < 0.05 was considered to indicate a significant difference. For all comparisons and tests, a two-sided critical significance level (p) was used.

## 3. Results

A total of 35 CAD patients and 43 control individuals participated in this study. The baseline characteristics of the CAD patients and controls are presented in Table 1. As can be seen in the table, there were no significant differences in age or gender distribution between these groups. In the CAD group, incidences of risk factors such as hypertension and DM were significantly higher than in the control group.

Table 2 presents results of the clinical and laboratory blood tests of patients in the pre- and postoperative periods of CABG. In the postoperative period, patients experienced decreases in platelets (PLT), fibrinogen, PATT, RBC, and Hb, but the level of hematocrit (HCT) remained unchanged, while WBC rose significantly. Inflammatory markers (ferritin, interleukin-6 (IL-6), and C-reactive protein (CRP)) also increased significantly.

Plasma aminothiol and blood GSH levels are presented in Table 3. Compared to controls, CAD patients were characterized, on admission, by decreased plasma levels of tCG and SAM and a decreased SAM/SAH ratio, as well as increased levels of other plasma thiols (tCys, tGSH, and tHcy). Bypass surgery did not have a significant effect on these parameters. The level of bGSH and its RS in patients was lower than that in controls, and after surgery it became even lower. However, the ratio of bGSH/Hb in patients after CABG, by contrast, increased in comparison to the baseline values. CABG had no significant effect on blood RS GSH.

Significant differences in the effect of plasma aminothiols on the blood GSH pool were found between patients and controls (Table 4). Thus, the control group was characterized by a rather strong positive correlation of GSSG with bGSH (Figure 1A). This association was not observed before CABG, but appeared after surgery (Figure 1B,C). As can be seen in Figure 1C, patients with high levels of GSSG also had higher levels of bGSH in the postoperative period. We therefore divided this group into two subgroups using the cut-off criterion GSSG = 4 μM (subgroups 1 and 2; see Figure 1C). A comparative analysis showed that subgroup 2 patients were characterized not only by a decrease in RS GSH, despite an increased level of bGSH in the postoperative period, but also by an increased level of blood glucose in the preoperative period (Figure 1D–F). We found no significant differences in preoperative aminothiol levels between these subgroups, with the exception of tGSH, which was slightly lower in patients in subgroup 2 (Figure 1G). The frequency of DM in these subgroups was statistically the same (55 vs. 46%, *p* > 0.05).

The control group and patients in the preoperative period were characterized by a negative correlation between bGSH and tCys. In addition, controls showed a negative association between SAH (Hcy precursor) and bGSH, and patients showed a negative association between tHcy and bGSH. After the operation, these indicators did not demonstrate associations with each other.

The control group and patients before surgery showed different association patterns of plasma aminothiols with each other (Table 4). If, in the first case, only a negative relationship between tCys and tGSH was revealed, then the patients were characterized by a rather close association of tCys with other thiols (tCG, tHcy, SAM, and SAH). In addition, the level of tHcy was associated with its precursors (SAM and SAH) in patients. After CABG, all these associations disappeared, but a negative correlation between SAM and tGSH appeared.

We also conducted an additional analysis of the influences of such binary factors as the CABG technique, heredity, history of atrial fibrillation, stage of hypertension, NYHA functional class, atherogenic coefficient, history of infarction, diet, presence of DM, and smoking on levels of aminothiols in the pre- and postoperative periods using the Mann–Whitney test, but did not reveal significant differences in any of these criteria.

## 4. Discussion

### 4.1. Impacts of Aminothiols and Glucose on GSH Metabolism in CAD

In the pathogenesis of coronary atherosclerosis, the role of ferroptosis, a type of programmed cell death characteristic of different cell types, including endothelial cells, cardiomyocytes, and vascular smooth muscle cells (VSMCs), is being actively studied. Ferroptosis is accompanied by the accumulation of iron in cells and the generation of a high level of ROS, the main source of which is lipid peroxidation [34]. To date, a number of genes have been identified that play key roles in protection against apoptosis, two of which are closely associated with GSH metabolism. These anti-ferroptotic genes include glutathione peroxidase 4 (GPX4). A negative association of coronary atherosclerosis with GPX4 in humans was recently identified in [35]. By contrast, GPX4 overexpression reduced atherosclerotic vascular damage in mice [36]. This enzyme catalyzes the reduction of phospholipid hydroperoxides using GSH, which is then oxidized to GSSG. In turn, GSSG is reduced by glutathione reductase using NADPH [37]. Therefore, the balance of oxidized and reduced GSH in cells is in a state of dynamic equilibrium that is provided by both its oxidation/reduction reactions and its synthesis/hydrolysis/elimination reactions, and RS GSH can be used to evaluate the performance of a given system. The importance of the role of reduction of RS GSH in calcified and atherosclerotic vessels, as well as in the endothelial-to-mesenchymal transition, has been shown in HUVEC cultures and in vivo (in human aortic valve fibrosis) [38,39]. The molecular mechanism that links the decrease in RS GSH in cells with these pathological processes is believed to be based on the aberrant glutathionylation of intracellular proteins. In our study, we found that the level of bGSH and its RS in CAD patients were significantly reduced. This is generally consistent with the results of previous clinical studies [3,4,26,40]. The presence of a close association of GSSG with bGSH reflects the ability to finely regulate RS GSH in normal conditions. In addition to the previous results, our new data show that the absence of such an association in CAD patients suggests not only that CAD is accompanied by a transition of the GSH redox system to a more pro-oxidant level, but also that this dysregulation manifests itself to varying degrees. 

Currently, there are several clinically significant factors that, through NADPH, influence the activity of GPX4. The first factor is hyperglycemia. In our study, there was no significant effect of glucose on the level of bGSH, which, apparently, was due to both the low impact of fasting glucose and the small number of CAD patients that were examined. 

The simultaneous determination of plasma aminothiols revealed a negative association of tCys with bGSH (bGSH/Hb) that was specific to patients with CAD at admission and controls. To our knowledge, this link to CAD has not yet been described in the literature. Three amino acids are required for the formation of GSH: glutamate, Cys, and glycine. Cys is a rate-limiting substrate for GSH synthesis [13]. Based on this, a positive relationship between tCys and bGSH should be expected; however, the results indicate that the level of tCys does not reflect the potential for its use by cells for GSH synthesis. On the contrary, tCys is a marker of GSH depletion in cells.

The transport of cysteine into cells is essential for GSH synthesis. Therefore, in addition to GPX4, cystine/glutamate antiporter solute carrier family 7 member 11 (SLC7A11), a protein that is part of the antiporter (system xc-), is one of the important anti-ferroptotic genes. The content of cystine in plasma (~50 μM [41]) is significantly higher than that of the reduced form of Cys (~5 μM [14]), which makes it the most accessible physiological form of Cys for cells.

It was found that the suppression of SLC7A11 expression reduced the content of GSH and induced ferroptosis in VSMC cultures [42]. In the same work, it was shown that direct GSH depletion significantly promoted the calcification of VSMCs. Thus, the inhibition of xc- appears to be the leading factor in intracellular Cys deficiency, leading to GSH pool depletion and, consequently, the activation of lipid peroxidation. By comparison, overexpression of SLC7A11 suppressed the ferroptosis of cardiomyocytes [43]. Recently, a positive effect of SLC7A11 mRNA methylation on the expression of this gene was revealed [44]. This suggests that a decrease in the level of SAM, as a donor of methyl groups, in patients with CAD may also be involved in the regulation of xc-.

Reduced Cys can enter cells via neutral amino acid transporters such as SLC1A4/5, SLC7A5, and SLC7A10; however, it is not yet clear whether they play a significant role in GSH homeostasis in cardiac atherosclerosis. For example, so far it has only been shown that SLC7A5 expression is essential for the de novo synthesis of GSH in trophoblast cultures [45]. A significant effect of SLC7A10 on GSH in adipocytes was also found [46].

The close correlation of tHcy and tCys levels in preoperative patients indicates that Hcy is an important endogenous source of Cys formation. The Hcy transsulfuration pathway also makes a significant contribution to the replenishment of the cellular Cys pool, the intensity of which should increase under HHcy conditions due to the fact that the optimum functioning of cystathionine-β-synthase (the rate-limiting enzyme of this pathway) is achieved only with a high content of Hcy in the cell [47]. It is interesting to note that, according to our results, although there was no significant effect of tHcy on bGSH under normal conditions, there was a rather strong negative correlation between the level of SAH, the precursor of Hcy, and bGSH in the controls. At the same time, an equally significant negative association between tHcy and bGSH in CAD patients indicates a negative role of Hcy in GSH metabolism. Apparently, this is due to the fact that Hcy downregulates the expression of cystathionine-β-synthase in cardiomyocytes [48], which causes the dysregulation of H2S metabolism [49,50]. In addition, it is believed that the direct binding of Hcy to H2S also contributes to a decrease in the bioavailability of the latter [51]. Because H2S increases GSH levels by enhancing the activity of γ-glutamylcysteine synthetase and up-regulating cystine transport [52], and H2S also inhibits various pro-oxidant effects of Hcy (the activation of NADPH oxidases, the uncoupling of endothelial NO synthase, and the inhibition of superoxide dismutase) [50], it becomes clear why HHcy ultimately has a negative impact on GSH. 

Schematically, the main changes in aminothiols in CAD are presented in Figure 2A.

### 4.2. Impact of CABG on Aminothiols

In the present study, we found that CABG induces a number of changes in blood GSH homeostasis. In the pathophysiological conditions associated with OS, there is an overall increase in GSH demand for antioxidant reactions, conjugation reactions, and the reduction of protein disulfides [37]. Thus, an increase in bGSH was observed in the early postoperative period after on-pump CABG [6]. However, in contrast to the previous study, we did not find an increase in the level of bGSH after CABG; on the contrary, we observed a decrease in this indicator. More than 99% of the bGSH pool is concentrated in RBCs; therefore, after massive blood loss (more than 10% of the circulating blood volume), hemodilution occurs and the level of bGSH should decrease. At the same time, our study revealed an increase in the concentration of GSH in relation to the concentration of Hb in the postoperative period, which is, apparently, the body’s response to blood loss, damage to chest tissues, and cardiac ischemia–reperfusion during CABG. This is also consistent with the results of [53], where an increase in the concentration of GSH in erythrocytes was found seven days after CABG. At the same time, however, this indicator remained lower than in the control group, and in patients with acute myocardial infarction, the level of bGSH in RBCs was higher than in healthy volunteers [54]. As in [6], the authors believe that OS mechanisms play a key role in the activation of GSH synthesis.

It should be noted that according to our results, CABG did not have any significant effect on GSSG or RS GSH in the entire patient cohort. However, the association of the GSSG and bGSH levels became significant in the postoperative period, which can also be interpreted as the activation of GSH synthesis in response to increased oxidation. We found that after CABG, about a third of the patients had increases in their GSSG levels of more than 4 µM. Despite the fact that the level of bGSH in this subgroup also increased, RS GSH was significantly lower than in the other patients, which clearly indicates sensitivity in these patients to the pro-oxidant effect of CABG. However, we found that preoperative glucose levels were significantly higher in those patients who had conditionally high GSSG and low RS GSH after CABG. In other words, elevated glucose levels negatively affect blood RS GSH specifically in the postoperative period, as shown schematically in Figure 2B. In addition, it is interesting to note that these patients had slightly lower levels of tGSH in the preoperative period than the rest of the patients. Whether tGSH can be used as a predictor of RS GSH imbalance in CABG remains an open question, since no significant association of this parameter with the cellular GSH pool or glucose levels was found, either before or after CABG.

An increase in the level of glucose in the cell leads to an increase in the consumption of NADPH in the reaction of reducing glucose to sorbitol, which is catalyzed by aldose reductase. It was shown that hyperglycemia exaggerates ischemic reperfusion and myocardial ferroptosis, which is also due to the activation of NADPH oxidase 2, and that the inhibition of this enzyme attenuates this hyperglycemic effect [55]. In addition, it has been established that hyperglycemia induces ferroptosis along the p53/xc-/GSH axis [56].

Thus, our results indicate the relevance of using various approaches to correct the GSH level in order to reduce the risk of postoperative complications of CABG (both through control of glucose levels and through intake of GSH precursors N-acetylcysteine and glycine) in patients with elevated glucose levels.

Data on the effect of CABG on plasma aminothiols are incomplete and inconsistent. We were unable to find information on changes in tCys, tCG, and tGSH in systemic circulation during the early postoperative period of CABG. It was found that from the first minutes of reperfusion and on the first day after surgery, the level of reduced GSH in the plasma significantly decreases [57], but it was not known how this indicator changes in the subsequent period. Significant increases in GSSG in blood/plasma and GSH-Cys disulfide in plasma were also found in the first 30 min of reperfusion in cold intermittent blood cardioplegia [58,59]. These data indicate that the greatest changes in the levels of plasma aminothiols in the postoperative period occur in the first hours and that their levels stabilize over several days.

Although our study did not find significant changes in plasma aminothiol composition after CABG, it was found that the administration of CABG had a dramatic effect on the relationships of aminothiols to each other and bGSH. Thus, there was a loss of reliability in the positive correlation of the SAM/SAH/Hcy/Cys axis in the postoperative period. The presence of a positive association suggests that the plasma pool of these compounds reflects the relationship of their cellular metabolism and the role of regulatory factors in the methionine cycle, and the transsulfuration pathway in particular. However, the plasma pool of thiols is also affected by other factors (transmembrane transport, excretion by the kidneys, hydrolysis, oxidation, transamination, desulfurization, etc.), which can significantly distort these bonds.

The only significant association within the plasma pool of analytes that was found in the postoperative period was a negative correlation between SAM and tGSH levels. Since SAM is a methyl group donor, a positive association between it and tGSH should be expected. SAM previously demonstrated the ability to attenuate decreases in GSH levels by preventing γ-glutamyltransferase activation and the expression of γ-glutamylcysteine ligase [60,61] It is interesting to note that a similar result was previously found in patients with coronavirus infections [62]. However, its negative character indicates that an increase in the level of SAM may reflect the inhibition of methyltransferase reactions, as has already been shown, for example, in a model of endotoxic shock [63].

A limitation of this study lies in the relatively low number of patients, which is due to the exploratory nature of the study; the number of study patients could not be determined based on statistical methods because the distribution parameters of aminothiols in the population as a whole are unknown. Second, differences in micronutrient intake supplements (vitamins B, methionine and others) and dietary intergroup differences may also have influenced the results. Third, the present study did not analyze patients for methionine cycle gene polymorphisms and transsulfurization pathways that are associated with HHcy. Although the heterogeneity of the groups according to this criterion is unlikely, it should not be excluded either.

## 5. Conclusions

Patients with CAD are characterized not only by the depletion of the bGSH pool but also by its redox status. The negative association of tCys and tHcy with bGSH points to the important role of reducing the bioavailability of the extracellular pools of Cys and Hcy, impacting Cys- and Hcy-dependent mechanisms of OS in GSH homeostasis. The present study indicates that CABG causes disruptions in aminothiol metabolism and induces synthesis of glutathione in erythrocytes, although CABG does not have a significant effect on the total aminothiol content. Moreover, glucose becomes an important factor in the dysregulation of GSH metabolism in CABG. The clinical implications of these findings need to be investigated.

## Figures and Tables

**Figure 1 metabolites-13-00743-f001:**
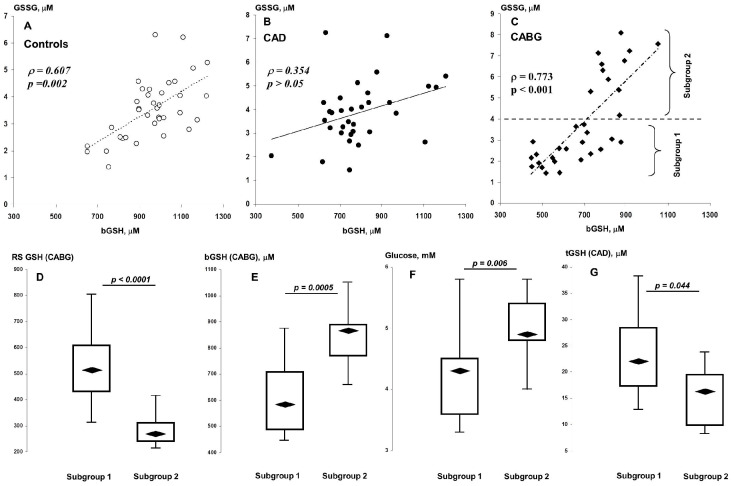
Associations of bGSH with GSSG in controls (**A**), CAD (**B**), and CABG (**C**) patients. Distributions of postoperative RS GSH (**D**) and bGSH (**E**) and preoperative glucose (**F**) and tGSH (**G**) in subgroups 1 (“low GSSG”) and 2 (“high GSSG”).

**Figure 2 metabolites-13-00743-f002:**
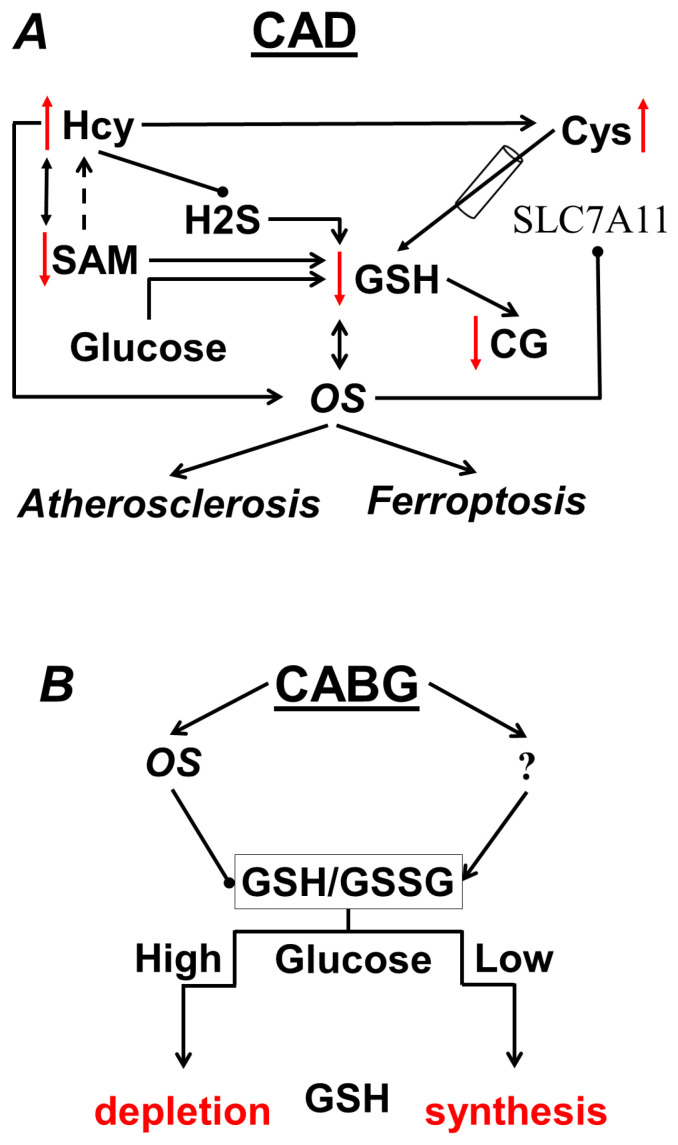
(**A**) Changes in the aminothiol system characteristic of CAD; (**B**) effect of glucose on GSH after CABG.

**Table 1 metabolites-13-00743-t001:** General characteristics of the study participants.

	CAD Group (N = 35)	Control Group (N = 43)	*p*
Sex: M (%)/F (%)	23 (66%)/12 (34%)	22 (51%)/21 (49%)	NS ^#^
Age (y)	59 [55; 63.5]	57 [52; 63.5]	NS *
CABG	On-pump: 10 (29%) Off-pump: 25 (71%)	-	
Hypertension: Yes (%)/No (%)	35 (100%)/0 (0%) 2nd stage: 10, 3rd stage: 25	0 (0%)/43 (100%)	<0.001 ^#^
DM: Yes (%)/No (%)	17 (49%)/18 (51%)	0 (0%)/43 (100%)	<0.001 ^#^
Fibrillation: Yes (%)/No (%)	21 (60%)/14 (40%)	0 (0%)/43 (100%)	<0.001 ^#^
Infarct: Yes (%)/No (%)	25 (71%)/10 (29%)	0 (0%)/43 (100%)	<0.001 ^#^
Smoke: Yes (%)/No (%)	21 (60%)/14 (40%)	25 (58%)/18 (42%)	NS ^#^
Diet ^1^: Yes (%)/No (%)	8 (23%)/27 (77%)	10 (23%)/33 (77%)	NS ^#^
Blood loss (mL)	650 (min 500, max 700)	-	
Functional class according to NYHA ^$^	II: 23 (66%) III: 12 (34%)	-	
Atherogenic coefficient		-	
Normal (<3.5)	16 (46%)		
Elevated (>4)	19 (54%)		
Left ventricular ejection fraction (%)	55 [52; 58]	-	
Obesity	4 (11%)	-	
Hyperlipidemia	2 (6%)	-	
Glucose (mM)	4.5 [4.1; 4.9]	-	
Cholesterol (mM)	4.4 [3.65; 4.85]	-	
LDL Cholesterol (mM)	2.5 [2.2; 3.1]	-	
HDL Cholesterol (mM)	1.1 [0.9; 1.4]	-	
HHcy (tHcy > 15 mkM)	27 (77%)	3 (7%)	<0.001 ^#^

* Mann–Whitney test; ^#^ binomial distribution test; ^$^ New York Heart Association; ^1^ According to the American Heart Association and the American Diabetes Association, the criteria for dietary compliance were limiting cholesterol intake to 200 mg/day and limiting the proportion of saturated fat in the daily food intake to 7%.

**Table 2 metabolites-13-00743-t002:** Laboratory parameters in CAD patients.

Indicator	Preoperative Level	Postoperative Level	*p* ^a^
HCT (%)	40 [38; 41]	40 [38; 40]	NS
RBC (10^6^/μL)	4.9 [4.7; 5.3]	3.8 [3.7; 3.9]	**<0.001**
Hb (g/L)	140 [136; 145]	99 [93; 104]	**<0.001**
MCV (fL)	89 [87; 96.5]	92 [88; 97.5]	NS
WBC (10^3^/μL)	7 [5.5; 7.0]	20 [16; 22]	**<0.001**
PLT (10^3^/μL)	259 [211; 311]	154 [127; 195]	**<0.001**
PATT (sec)	32 [29; 34]	26 [25; 28]	**<0.001**
CRP (mg/L)	4 [3; 6]	24 [20; 34.5]	**<0.001**
Fibrinogen (g/L)	3.9 [3.7; 4.1]	2.7 [2.3; 3.0]	**<0.001**
Ferritin (μg/L)	68 [49; 83.5]	190 [169; 295]	**<0.001**
IL-6 (pg/mL)	5 [3; 6]	19 [14; 24]	**<0.001**

^a^ Between CABG pre- and postoperative periods (paired Wilcoxon rank test).

**Table 3 metabolites-13-00743-t003:** Changes in aminothiol levels in patients in the CAD group and in the control group.

Indicator	Controls	*p* ^a^	Preoperative Level	*p* ^b^	Postoperative Level
tCys (μM)	286 [277; 299]	**<0.001**	331 [294; 368]	NS	290 [244; 330]
tCG (μM)	31.1 [27.5; 35.4]	**<0.001**	23.5 [21; 29]	NS	24.7 [22.1; 29.8]
tGSH (μM)	10.8 [8.6; 11.9]	**<0.001**	20.5 [16.6; 23.8]	NS	16.6 [13.2; 19.0]
tHcy (μM)	12.3 [11.6; 13.3]	**<0.001**	18.2 [15.3; 22.2]	NS	17.9 [14.4; 20.4]
SAM (nM)	104 [91; 119]	**<0.001**	72 [62; 94]	NS	79 [73; 96]
SAH (nM)	9.4 [7.3; 11.6]	NS	11.1 [7.1; 15.7]	NS	15.2 [9.4; 17.9]
SAM/SAH	11.0 [9.0; 13.2]	**<0.001**	6.0 [4.0; 9.2]	NS	5.8 [4.5; 7.8]
bGSH (μM)	968 [876; 1028]	**<0.001**	755 [682; 842]	**0.005**	698 [554; 802]
GSSG (μM)	3.54 [2.80; 4.23]	>0.05	3.9 [3.1; 4.8]	NS	3.0 [2.2; 6.5]
RS GSH	545 [466; 647]	**<0.001**	408 [339; 484]	NS	416 [269; 532]
bGSH/tCys	3.41 [2.93; 3.70]	**<0.001**	2.36 [1.93; 2.64]	NS	2.39 [1.93; 2.90]
bGSH/Hb (μmol/g)	-	-	5.45 [4.74; 6.04]	**<0.001**	7.21 [5.55; 8.34]

^a^ Between controls and CABG preoperative period (Mann–Whitney rank test); ^b^ between CABG pre- and postoperative periods (paired Wilcoxon rank test).

**Table 4 metabolites-13-00743-t004:** Associations of aminothiol levels in patients and controls.

Indicators	Controls		CAD		CABG	
	R *	*p*	R *	*p*	R *	*p*
bGSH and GSSG	**0.607**	**0.002**	0.354	NS	**0.773**	**<0.001**
bGSH and tCys	**−0.538**	**0.011**	**−0.429**	**0.043**	0.097	NS
bGSH/Hb and tCys	-	-	**−0.415**	**0.045**	0.091	NS
bGSH and tHcy	0.217	NS	**−0.545**	**0.018**	0.006	NS
bGSH/Hb and tHcy	-	-	**−0.531**	**0.025**	0.093	NS
bGSH and SAH	**−0.563**	**0.005**	−0.078	NS	0.225	NS
GSSG and tCys	**−0.551**	**0.011**	−0.275	NS	0.213	NS
tCys and tCG	0.372	NS	**0.505**	**0.043**	0.354	>0.05
tCys and tHcy	0.214	NS	**0.743**	**<0.001**	0.354	NS
tCys and SAM	0.049	NS	**0.707**	**<0.001**	−0.045	NS
tCys and SAH	0.199	NS	**0.565**	**0.018**	0.117	NS
tCys and tGSH	**−0.484**	**0.028**	0.150	NS	0.067	NS
tHcy and SAM	−0.27	NS	**0.546**	**0.019**	0.375	NS
tHcy and SAH	−0.078	NS	**0.664**	**0.001**	0.462	NS
tGSH and SAM	−0.181	NS	0.218	NS	**−0.522**	**0.042**

* Pearson correlation coefficient.

## Data Availability

Data will be made available upon reasonable request by the corresponding author. Data are not publicly available due to privacy or ethical restrictions.

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
