# Peer review of "Influence of Coronary Artery Bypass Grafts on Blood Aminothiols in Patients with Coronary Artery Disease"

_metabolites, 2023, doi:10.3390/metabo13060743_

Round 1
Reviewer 1 Report
The manuscript by Alexander Vladimirovich Ivanov et al. investigate the associations of blood (b) GSH with glucose and plasma aminothiols (cysteine, homocysteine, cysteinyl-glycine, S-adenosylmethionine and S-adenosylhomocysteine) in patients with coronary artery disease (CAD) before coronary artery bypass graft (CABG) procedure and in the early (4-5 days) post-operational period.
While the topic is interesting, it is certainly not original.
There are several points that need to be addressed:
1. Introduction paragraph, although well described, it is too long and needs to be shortened.
2. I suggest to the authors to eliminate the word "blood" before "plasma" in the paragraph Materials and Methods, lines 178, 179, 196, 197….. because it creates confusion.
3. The study cohort is really small. Did the authors calculate the sample size? If no, why?
4. How did the authors define the diet (Table 1)?
5. The atherogenic coefficient needs to be defined, as well as the risk factors. How “normal” or “elevated” atherogenic coefficient means?
6. In table 2, among different routine parameters, there is also IL-6. This is not a molecule, usually analyzed during standard biochemical assessment. How did the authors dose it?
7. There are many typos, especially in the measurement units like: Table 1 last line tHcy>15 mkM; line 257 (GSSG = 4 M)
8. In the discussion section the authors may introduced the study limitation, including first of all the small size of sample.
I suggest the authors a moderate editing of English language
Author Response
We thank you and the reviewers for their attention to our work. We have revised our manuscript in accordance with the comments of the reviewers, thanks to which, as we believe, it was possible to improve the work. You can see all the highlighted text changes in the attached file (highlighted manuscript.docx).
The manuscript by Alexander Vladimirovich Ivanov et al. investigate the associations of blood (b) GSH with glucose and plasma aminothiols (cysteine, homocysteine, cysteinyl-glycine, S-adenosylmethionine and S-adenosylhomocysteine) in patients with coronary artery disease (CAD) before coronary artery bypass graft (CABG) procedure and in the early (4-5 days) post-operational period.
While the topic is interesting, it is certainly not original.
There are several points that need to be addressed:
- Introduction paragraph, although well described, it is too long and needs to be shortened.
Ans.: the Introduction section has been shortened. At the same time, a number of additions were made to this section in order to more clearly present the relevance of the study. - I suggest to the authors to eliminate the word "blood" before "plasma" in the paragraph Materials and Methods, lines 178, 179, 196, 197….. because it creates confusion.
Ans.: the text has been corrected according to the comment.
- The study cohort is really small. Did the authors calculate the sample size? If no, why?
Ans.: determination of the required sample size is a very important issue, however, due to the exploratory nature of our study and the lack of data on the parameters of the distribution of analytes in the population, we could not calculate the minimum sample before the start of the study. In fact, we initially conducted the analysis using only a sample of 20 patients, and after receiving the first data, we expanded this sample to 35 people. Comparing the parameters of this sample with a sample of controls close in volume, we used the group comparison method to conclude that this volume should be sufficient for most aminothiols. It should be noted here that when comparing groups on indicators such as SAH or GSSG, several hundred or even thousands of patients would be needed to identify significant differences, which makes it impossible to complete the study in a reasonable time.
- How did the authors define the diet (Table 1)?
Ans.: Table 1 has been supplemented with a comment: “1 According to the American Heart Association and the American Diabetes Association, the criteria for dietary com-pliance were limiting cholesterol intake to 200 mg/day and limiting the proportion of saturated fat in the daily food intake to 7%”.
- The atherogenic coefficient needs to be defined, as well as the risk factors. How “normal” or “elevated” atherogenic coefficient means?
Ans.: According to [Klimov A.N., Nikulcheva N.G. Lipid and lipoprotein metabolism and its disorders. —St. Petersburg: Peter Com, 1999 - 512 p.] the normal atherogenic index is 2.2-3.5 mM. An increased coefficient - more than 4 mM reveals a high probability of atherosclerotic vascular damage. A note has been added to Table 1.
- In table 2, among different routine parameters, there is also IL-6. This is not a molecule, usually analyzed during standard biochemical assessment. How did the authors dose it?
Ans.: The following text has been added to the Methods section: “The determination of IL-6 in plasma was carried out using test systems from Bender Medsystems GmbH (Vienna, Austria) according to the manufacturer's instruction [31].”
- There are many typos, especially in the measurement units like: Table 1 last line tHcy>15 mkM; line 257 (GSSG = 4 M)
: typos have been corrected.
- In the discussion section the authors may introduced the study limitation, including first of all the small size of sample.
Ans.: The following text has been added to the discussion section: “A limitation of this study lies on the relatively low number of patients, which is due to the exploratory nature of the study and number of study patients could not be deter-mined based on statistical methods because the distribution parameters of aminothiols in the population as a whole are unknown. Second, differences in micronutrient intake supplements (vitamins B, methionine and other) and dietary intergroup differences may also have influenced the results. Third, the present study did not analyze patients for methionine cycle gene polymorphisms and transsulfurization pathways that are associated with HHcy. Although the heterogeneity of the groups according to this criterion is unlikely, it should not be excluded either.”
Comments on the Quality of English Language
I suggest the authors a moderate editing of English language
Ans.: our manuscript has been edited of English language by MDPI (certificate attached). If, in the opinion of the reviewer, the manuscript still needs editing, then we can re-send it for revision.

Reviewer 2 Report
This clinical study in patients with coronary artery disease (CAD) undergoing coronary artery bypass graft (CABG) surgery examines whether there is a correlation between blood glutathione levels (bGSH) and glucose and plasma aminothiols and whether these associations are influenced by the surgery. It was shown that bGSH had a negative correlation with aminothiols prior to surgery but this was changed to no correlation during and after surgery. CABG caused a disruption of the bGSH metabolism, with glucose levels having an involvement.
This would appear to be an observational study looking at various metabolic factors associated with CABG and the oxidative stress that occurs during cardiac surgery. It is noted that cardiopulmonary bypass appears to influence this more than when surgery is conducted off-pump, but the investigators chose to include both these techniques in their study cohort. No subgroup analysis has been conducted! One wonders whether this fact alters the outcome of this study! There appears to be no attempt to decide whether manipulation of the metabolic factors could be a useful therapeutic option during CABG surgery. The manuscript is very long and would benefit from being shortened, especially the Introduction and Discussion.
Specific Comments.
Abstract:
· Is it more correct to state that CAD and CABG is associated with a decrease in bGSH rather than the other way round? It is unlikely that decreased bGSH results in CAD or CABG!
· What is the rationale for thinking that glucose and plasma aminothiols are associated with bGSH?
· What is meant by the statement "CABG had no significant effect on these parameters, with the exception of an increase in the bGSH/hemoglobin ratio " Is this at admission as implied by previous sentence?
· Is 'associations' the correct term or should it be 'correlations'?
· What are the implications of this study for CABG and postoperative outcome?
Introduction:
· What other aminothiols are present?
· What is the hypothesis of this study? What is the objective of measuring these metabolites? Do the authors think that there can be a therapeutic indication for being able to manipulate these levels?
Methods:
· What is meant by 'the main coronary arteries'?
Results:
· Table 1 - Avoid using p>0.05 as it can be confused for a significant difference! Either state NS or leave blank!
· Table 1 - The introduction indicates that differences can be observed with on-pump and off-pump surgery, so why muddy the waters by including both in the CAD group? Are these 2 techniques sub-group analysed? This would appear to be a flaw in the study as all postoperative values have been combined!
· Why was the cut-off criterion of 4µM selected? Is there a rationale for this?
Discussion:
· Discussion is long and could be shortened to make the argument clearer. Perhaps a diagram to show the relationship of all the metabolic factors involved would aid clarity.
Author Response
We thank you and the reviewers for their attention to our work. We have revised our manuscript in accordance with the comments of the reviewers, thanks to which, as we believe, it was possible to improve the work. You can see all the highlighted text changes in the attached file.
This clinical study in patients with coronary artery disease (CAD) undergoing coronary artery bypass graft (CABG) surgery examines whether there is a correlation between blood glutathione levels (bGSH) and glucose and plasma aminothiols and whether these associations are influenced by the surgery. It was shown that bGSH had a negative correlation with aminothiols prior to surgery but this was changed to no correlation during and after surgery. CABG caused a disruption of the bGSH metabolism, with glucose levels having an involvement.
This would appear to be an observational study looking at various metabolic factors associated with CABG and the oxidative stress that occurs during cardiac surgery. It is noted that cardiopulmonary bypass appears to influence this more than when surgery is conducted off-pump, but the investigators chose to include both these techniques in their study cohort. No subgroup analysis has been conducted! One wonders whether this fact alters the outcome of this study! There appears to be no attempt to decide whether manipulation of the metabolic factors could be a useful therapeutic option during CABG surgery. The manuscript is very long and would benefit from being shortened, especially the Introduction and Discussion.
Specific Comments.
Abstract:
- Is it more correct to state that CAD and CABG is associated with a decrease in bGSH rather than the other way round? It is unlikely that decreased bGSH results in CAD or CABG!
Ans.: text has been corrected: “Coronary artery disease (CAD) and the coronary artery bypass graft (CABG) are associated with a decreased blood glutathione (bGSH) level.”
- What is the rationale for thinking that glucose and plasma aminothiols are associated with bGSH?
Ans.: the following text has been added: «Since GSH metabolism is closely related to other aminothiols (homocysteine and cysteine) and glucose, the aim of this study was to reveal the associations of bGSH with glucose and plasma aminothiols in CAD patients (N=35) before CABG and in the early postoperational period.”
- What is meant by the statement "CABG had no significant effect on these parameters, with the exception of an increase in the bGSH/hemoglobin ratio " Is this at admission as implied by previous sentence?
Ans.: Yes, that is right.
- Is 'associations' the correct term or should it be 'correlations'?
Ans.: we have used the term 'associations' in a broad sense, meaning both the 'correlations' of metabolites with each other, and the decrease or increase in their levels under certain conditions
- What are the implications of this study for CABG and postoperative outcome?
Ans.: this is a very interesting question, but it goes beyond the scope of this study. We are currently only going to analyze the postoperative outcome in these patients.
Introduction:
- What other aminothiols are present?
Ans.: aminothiols include a large number of compounds, but the main ones present in organisms include glutathione, cysteine, homocysteine, cysteinylglycine and gamma-glutamylcysteine
What is the hypothesis of this study? What is the objective of measuring these metabolites? Do the authors think that there can be a therapeutic indication for being able to manipulate these levels?
Ans.: text has been added: “In particular, nothing is known about the effect CABG has on aminothiols, primarily GSH in the blood early postoperative (4-5 days) period, and whether glucose levels may play a role here. It is also not known which associations of aminothiols with each other are characteristic of CAD and whether they are retained after CABG. To shed light on these questions we conducted a study of the total contents of total plasma aminothiols (tCys, tHcy, tCG, tGSH), Hcy precursors (SAM, and SAH) and reduced and oxidized GSH in the blood (bGSH and GSSG, respectively) of patients in the preoperative and early postoperative (4-5 days) CABG periods.”
Methods:
- What is meant by 'the main coronary arteries'?
Ans.: text has been added: “The main inclusion criteria for patients in the main group were the presence of hemodynamically significant stenosis of the main coronary arteries (the anterior interventricular artery, circumflex artery, right coronary artery), planned operation (surgical myocardial revascularization), an age of 45-75 years, and the presence of informed consent”
Results:
- Table 1 - Avoid using p>0.05 as it can be confused for a significant difference! Either state NS or leave blank!
Ans.: the text has been corrected.
- Table 1 - The introduction indicates that differences can be observed with on-pump and off-pump surgery, so why muddy the waters by including both in the CAD group? Are these 2 techniques sub-group analysed? This would appear to be a flaw in the study as all postoperative values have been combined!
Ans.: Initially, the study of the influence of operating techniques was one of the main objectives of this work. However, after analyzing the data, we did not find any significant effect of the surgical technique on thiol homeostasis, despite the fact that there were prerequisites for this in the literature. Obviously, to solve this problem, it is necessary to conduct a larger study. Therefore, we have indicated this very briefly in the results: “We also conducted an additional analysis of the influences of such binary factors as the CABG technique, heredity, the history of atrial fibrillation, the stage of hypertension, the NYHA functional class, the atherogenic coefficient, the history of infarction, the diet, the presence of DM, and smoking on the levels of aminothiols in the pre- and postoperative periods using the Mann–Whitney test, but did not reveal significant differences in any of these criteria (data not shown).”
- Why was the cut-off criterion of 4µM selected? Is there a rationale for this?
Ans.: the choice of this threshold value was due to its proximity to the median level of GSSG in patients in the preoperative period, which made it possible to divide this group into two subgroups approximately equal in number of patients.
Discussion:
- Discussion is long and could be shortened to make the argument clearer. Perhaps a diagram to show the relationship of all the metabolic factors involved would aid clarity.
Ans.: the Discussion section has been shortened but supplemented by Figure 2, where we schematically show what changes in aminothiols were characteristic of CAD (A) and the role of glucose in the metabolism of GSH after CABG (B).

Round 2
Reviewer 1 Report
The authors have clarified several of the questions I raised in my previous review. Only minor revisions are needed before it can be accepted:
1. In the “Introduction” paragraph:
Line 49 – delete “are available”
Line 80 – delete “in whole blood, this figure is referred to as”
Line 82 – Substitute “GSH” with “bGSH”
2. In the “Results” paragraph, Table 1:
Line 262 - <3.5 (mM?); >4 (mM?). The atherogenic coefficient (AC) is a ratio between parameters expresses with the same unit of measurement (total cholesterol – HDL/HDL). So I don’t understand the “mM”. How was the AC calculated?
Author Response
Dear Reviewer,
thank you very much for Your comments on our manuscript. We appreciate Your criticism and input into the manuscript as it helps to improve it. According to the Reviewers` suggestions, we have made some changes in the manuscript.
The authors have clarified several of the questions I raised in my previous review. Only minor revisions are needed before it can be accepted:
- In the “Introduction” paragraph:
Line 49 – delete “are available”
Ans.: this text was corrected: “Treatment of CAD includes two relatively invasive strategies that aim to re-establish an adequate blood supply to undersupplied myocardial territories due to severe coronary stenosis or vessel occlusion: percutaneous coronary intervention (PCI) and coronary artery bypass grafting (CABG) [1].”
Line 80 – delete “in whole blood, this figure is referred to as”
Ans.: this text was corrected: “Its intracellular concentration is very high (about 1 mM or more); moreover, ~99% of GSH is accounted for by the reduced form of GSH (bGSH).”
Line 82 – Substitute “GSH” with “bGSH”
Ans.: this text was corrected: “The ratio between the bGSH and the GSSG (redox status (RS)) is a recognized index of OS [12].”
- In the “Results” paragraph, Table 1:
Line 262 - <3.5 (mM?); >4 (mM?). The atherogenic coefficient (AC) is a ratio between parameters expresses with the same unit of measurement (total cholesterol – HDL/HDL). So I don’t understand the “mM”. How was the AC calculated?
Ans.: Yes, AC was calculated according according to the above formula. This error in table 1 has been corrected, “mM” was removed: “Atherogenic coefficient: Normal (<3.5) Elevated (>4)”
Best Regards,
authors

Reviewer 2 Report
The authors have done a good job of revising the manuscript riot in accordance with the comments and suggestions of this reviewer. The manuscript has been improved.
No concerns.
Author Response
Dear Reviewer,
we thank You for your attention to our work and for your valuable comments that allowed us to improve this manuscript.
Best Regards,
authors